# Effect of Aging on Healing Capacity of Bituminous Composites Containing Polyphosphoric Acid

**DOI:** 10.3390/ma16093333

**Published:** 2023-04-24

**Authors:** Mingxia Li, Elham Fini, Xiaomin Jia, Baiyang Song, Yanhong Wang

**Affiliations:** 1School of Civil Engineering, Luoyang Institute of Science and Technology, No. 90 Wangcheng Avenue, Luolong District, Luoyang 471023, China; lymzy1119@163.com (X.J.); baiysong@foxmail.com (B.S.); yhwanglla@126.com (Y.W.); 2School of Sustainable Engineering and Built Environment, Arizona State University, 660 S. College Avenue, Tempe, AZ 85287-300, USA; efini@asu.edu

**Keywords:** polyphosphoric acid, healing, modified bitumen, aging, ultraviolet

## Abstract

This study examines how aging affects the healing capacity of bituminous composites containing polyphosphoric acid (PPA). PPA is commonly used in bituminous composites to enhance its elasticity, however, PPA effectiveness highly depends on other constituents on the matrix and the environmental (internal and external) factors. In terms of internal factors, the interplay between PPA and various bitumen modifiers have been extensively studied. Here, we study how external factors such as exposure to ultraviolet radiation affect PPA’s efficacy, measured in terms of change in bitumen’s healing index. The study results showed that the introduction of PPA to bituminous composites significantly increases the bitumen healing index, however, the change in the healing index becomes less pronounced as aging progresses. The presence of additives such as taconite were found to affect the effect of PPA on bitumen’s healing index. For instance, bitumens containing 30% taconite showed the highest increase in their healing index in the presence of PPA among studied scenarios. Overall, bitumen containing PPA had a higher healing index than those without PPA regardless of the extent of aging and dosage of modifiers. This, in turn, indicates that PPA is highly effective for enhancing bitumen healing. This can be attributed to the role of PPA in promoting intermolecular interactions within the bitumen matrix.

## 1. Introduction

Bitumen is significantly used as a binder in mixtures for pavement material. During the service period, it is exposed to complex environmental conditions, such as ultraviolet (UV) oxidation. UV aging could trigger pavement destruction and seriously weakens the lifespan of bitumen pavement. Currently, the healing capacity is attracting more and more attention as a potentially helpful factor for promoting the lifespan of service [1]. As such, investigating and evaluating the effect of UV aging on the healing performance of bitumen will be helpful because it directly influences the service life of bitumen pavement [2]. In order to enhance asphalt self-healing, many scholars have explored the healing effects using different physical methods [3,4,5]. Some studies have attempted to use microcapsules, hollow-fiber tubes, and nanoparticles [6,7,8,9,10,11,12,13,14].

He et al. have studied the effect of thermos-oxidative aging on the healing capacity of bitumen. They found that a higher level of aging correlates to a smaller healing index. This difference heightened after multiple damage-healing cycles [15]. Xu and Qu et al. utilized a molecular dynamic simulation to investigate the effect of oxidative aging on healing capacity. The results showed that aging weakened the nano-aggregation behavior of asphaltene molecules and reduced the translational mobility of bitumen molecules. Compared to virgin bitumen, aged bitumen has a higher activation-energy barrier for self-healing [16,17]. Vallerga’s research indicated that UV radiation could apparently influence the softening point and ductility of bitumen [18]. Glotova et al. have investigated how UV aging affects bitumen by testing its chemical properties, structure, and composition contents before and after aging. They found that UV radiation could significantly degenerate the performance indicators above [19]. Wu et al. conducted a Fourier infrared test and a dynamic shear rheometer (DSR) test using a base bitumen and polymer-modified bitumen with a high-pressure mercury lamp. The results showed that UV radiation could obviously age bitumen, and different intensities were posed to different aging effects [20]. Although the researchers above indicated UV radiation can change the chemical contents and technical performance of bitumen material, they focused less on the UV-aging effect on the healing property of bitumen.

Polyphosphoric acid (PPA), as a modifying agent, has become the most important acid used in bitumen [21,22,23]. It can be produced by deriving H_3_PO_4_ from heating P_2_O_5_ or the dehydration of H_3_PO_4_ at high temperatures [24]. Introducing PPA to bitumen improves its high-temperature grade [25,26,27]. Edward et al. have analyzed the rheological performance of bitumen modified by PPA. They pointed out that PPA had a remarkable effect on the medium- and high-temperature rheological behavior of bitumen. Besides, PPA also promoted the resistance deformation utilizing a repeated creep test [28,29]. Zhang et al. investigated the structural characteristics of bitumen modified by PPA before and after short- or long-term thermal aging. The study showed that the suitable addition of PPA improved the major physical and rheological properties of SEBS-modified (SM) bitumen, and the improved properties became more obvious with further aging [30]. Also, they investigated the effect of PPA on the rheological behaviors of HVM bitumen before and after aging. The suitable gelation induced by PPA not only improved the high-temperature rheological performance of HVM bitumen but also increased the low-temperature rheological performance [31]. With the aim of improving the compatibility and high- and low-temperature properties of SBR-modified bitumen, Liang et al. studied the results of adding PPA to achieve a further modification. It showed that PPA remarkably enhanced the adhesion ability, high-temperature elasticity, and anti-rutting ability of modified bitumen [32]. Bennert and Martin researched the recyclability of PPA-modified bitumen. RAP (Reclaimed Asphalt Pavement) containing PPA is not detrimental to hot-mix bitumen and should perform in a similar manner to RAP containing SBS-modified bitumen [33]. Some studies also showed that when PPA-modified bitumen was used alone, the properties of bitumen and its mixture would be weakened. Thus, the PPA was often used in conjunction with other modifiers [34,35,36].

There are also studies that evaluated the effects of PPA and aging on bitumen performance. Zhang et al. found that a moderate gelation by 0.8 wt % PPA in styrene–ethylene/butylene–styrene-modified bitumen improved the softening point and elasticity before and after aging without negative effects [30].

Based on the lack of research on the healing capacity of bitumen in UV-aging environmental conditions, the current direction of research focuses on the effect of PPA on the self-healing ability of asphalt binder with taconite, investigated for the first time, and this paper also evaluates how external factors such as exposure to ultraviolet radiation affects PPA’s efficacy, measured in terms of change to the bitumen’s healing index. These are attributes that help us to understand how PPA improves capacities of bitumen holistically. Different UV-aging durations (0 h, 100 h and 200 h), with or without PPA, and different dosages of taconite were considered during the tests to evaluate effects on healing performance. Experiments were conducted using DSR equipment. The healing index was calculated according to the complex shear modulus. Also, this study analyzed the promoted effect of taconite on the healing performance of bitumen modified by PPA. It was found that ultraviolet (UV) radiation can weaken the healing capacity of bitumen modified by polyphosphoric acid and taconite. Inversely, polyphosphoric acid and taconite filler play a positive role on the healing capacity of modified bitumen.

## 2. Materials and Methods

### 2.1. Material and Sample Preparation

PG 64-22, which can be used in areas where the maximum temperature of the pavement is 64 °C and the minimum temperature is −22 °C, was used in this study. It was obtained from the Holy Frontier Corporation in Arizona, and its specific gravity is 1.041 g/cm^3^ at 15.6 °C and absolute viscosity is 179 Pa·s at 60 °C. Taconite was acquired from a Minnesota mine’s tailings. Based on the standard testing method [37,38,39], the values of specific gravity, abrasion resistance, and the dosage of deleterious materials were 2.803 g/cm^3^, <40, and 40–45%, respectively. Four types of bitumen specimens with different dosages of taconite and 1% PPA were tested (0%, 10%, 20%, 30%) (Figure 1). Polyphosphoric acid (83.3% P_2_O_5_, so-called 115% H_3_PO_4_) was acquired from Sigma-Aldrich. One percent wt % PPA was mixed in samples.

The PPA–taconite modified-binder samples were prepared using the high shear mixer. The mixing temperature, time, and mixing speed were 135 °C, 30 min, and 2000 rpm, respectively. Setting up the PPA-modified bitumen with taconite began with the low shear blend of 500 rpm for first 10 min, and following, the speed of blender was raised to 2000 rpm for a constant 20 min.

### 2.2. Ultraviolet (UV) Testing

In order to make sure the thick film was consistent, about 6 g of unaged sample was evenly poured onto a rubber mold. Then, the mold with the sample was placed into a UV chamber, which was a closed box with a UV-light source 10 cm away from the bottom. The testing temperature was 65 °C, the UV-radiation intensity was 0.71 W/m^2^, and the wavelength was 340 nm. The UV-aging time continued for 100 h and 200 h, respectively. Two replicates were conducted for each sample.

### 2.3. Healing Testing (Dynamic Shear Rheometer)

To evaluate the healing capacity of modified bitumen using PPA and taconite, a rheometry test was conducted utilizing a time-temperature sweep (loading-rest-loading) following ASTM-D7552-09 [40] by Anton Paar MCR302, whose maximum torque is about 230 mNm. A parallel plate set-up with 8 mm diameter and 2 mm thick samples were used at 25 °C. The test was performed for two replicates at a strain of 5% and a frequency of 10 Hz. All samples were loaded until its complex shear modulus (G^*^) was reduced to 50% of the initial value (G_0_), which was named G_a_. The loading was then stopped, and a rest period of 900 s was implemented. After the rest period, the loading was repeated until the complex shear modulus was reduced to 50% of G_b_, where G_b_ was the initial modulus after the rest period. Healing index was then calculated using Equation (1) following prior works [41,42,43].
(1)$$HI=\frac{G_{b}-G_{a}}{G_{0}-G_{a}}$$

G_0_—Initial dynamic shear modulus before rest period;

G_a_—Ending dynamic shear modulus before rest period;

G_b_—Initial dynamic shear modulus after rest period.

## 3. Results and Discussion

### 3.1. Effect of UV Aging on the Healing Capacity of Modified Bitumen

Figure 2 presents the healing index of bitumen modified by PPA and taconite samples before and after different UV-aging times. As shown in Figure 2, there were obviously differences among the healing indexes of unaged, 100 h UV-aged, and 200 h UV-aged samples. The healing indexes were weakened when the UV-aging period increased. It is suggested that the effect of UV radiation on the healing capacity has a remarkable influence. This may due to a dramatic increase in the stiffness and the loss of viscoelasticity after 20 h of UV exposure [44]. With a longer aging time, there is a larger amount of asphaltene in the asphalt. This makes the modified binders harder and, therefore, less able to flow at the same temperature, impairing its healing ability.

For unaged and 100 h UV-aged samples, the healing index was improved gradually with the increase of the dosage of taconite included. When the aging period was 200 h, the healing index was reduced. To further understand the effect of UV radiation on the healing performance of modified bitumen, the delta healing index of 0–100 h and 100–200 h UV-aging periods were calculated (Figure 3). Regardless of whether PPA was included, it showed that the healing index of the samples exposed to 0–100 h UV radiation were weakened more than that of the samples exposed to 100–200 h, besides the samples with 30% taconite content during the 100–200 h aging period. This indicates that the healing index decreased slowly with an increase in the aging time. During the 0–100 h aging period, those differences, which were reduced by 18.31% and 7.68% for without and with PPA, were the highest while the dosage of taconite was 10%, respectively. However, during the 100–200 h aging period, the highest difference, which was reduced by 4.08% and 20.16%, respectively, occurred while the dosage of taconite was 30%. It can be concluded that the healing capacity would be greatly destroyed when the UV-aging time was 200 h, even though it had 30% taconite and 1% PPA.

### 3.2. Effect of PPA on the Healing Capacity of Bitumen with Taconite

Figure 4 presents the healing indexes of unaged (a), 100 h aged (b), and 200 h aged (c) samples modified with or without PPA. As shown in Figure 4, the healing index was promoted after adding PPA to bitumen with taconite regardless of the UV-aging time. Due to a moderate dosage of PPA gelation, the rheological behavior of bitumen was improved [45]. The addition of PPA can increase the viscosity and adhesion ability of bitumen [31,32]. It is also helpful in enhancing the resistance of deformation, improving the self-healing property.

After 100 h of UV aging, the healing index of samples with PPA was much higher than that of samples without PPA, which were 34.87%, 35.16%, 38.95%, and 44.84%, while the dosage of taconite was 0%, 10%, 20%, and 30%, respectively. This indicates that PPA has a very powerful promotion effect on the healing property of bitumen. When exposed to 200 h of UV aging, it was found that the healing index of samples with PPA decreased with an increase in the taconite dosage. However, they were still a little higher than that of samples without PPA. To evaluate the effect of PPA on the healing performance of aged bitumen with taconite, the delta healing index, which represents the difference value of the healing indexes between taconite-modified binders without and with PPA, were computed (Figure 5). Regardless the type of UV aging, the delta healing indexes were almost all positive. While the UV-aging time was 100 h, the delta healing indexes were 18.47%, 16.47%, 16.45%, and 14.68%, respectively. With the increase in the taconite dosage, thee delta healing indexes decreased. When the UV-aging time was 200 h, the promotion of PPA on the healing property was gradually weakened. This was due to the healing property of bitumen being destroyed when the UV-aging time was 200 h. This result is consistent to that concluded in Section 3.1.

### 3.3. Effect of Taconite on the Healing Capacity of Bitumen Modified by PPA

Figure 6 shows bitumen modified by PPA with different dosages (0, 10%, 20%, and 30%) of taconite samples before and after UV aging. It can be observed that there are remarkable differences among healing indexes of the four kinds of samples. For unaged samples, the healing index was improved by increasing the taconite dosage. When included, 0%, 10%, 20%, and 30% dosages of taconite yielded healing indices of 36.88, 42.84, 45.32, and 46.81%, respectively. After conducting 100 h of UV aging, the healing capacity still showed a promoting trend; the values were 34.87, 35.61, 38.95, and 44.84%, respectively. This indicates that taconite can enhance the healing performance of bitumen due to the role of taconite in promoting its thermal conductivity, which is consistent with the results of our published paper [46]. Higher thermal conductivity can make the temperature of binders increase more quickly and reach a critical point that drives the asphalt binder to start the healing behavior.

However, when handled with 200 h of UV aging, the healing property was reduced, and with the addition of taconite, it increased. The delta healing indexes of different taconite dosages were also calculated (Figure 7). It can be seen that the delta healing index was increased with the dosage of taconite. When concentrations of 10%, 20%, and 30% taconite were added, they were increased by 5.96, 8.44, and 9.93% for unaged samples, respectively. After 100 h of aging, the delta healing indexes deceased. However, after 200 h of UV aging, the indexes were reduced significantly, and the values were 0.03, −2.71, and −9.9%, respectively. Hence, introducing taconite can enhance the healing performance of modified bitumen with PPA. However, when it was aged significantly longer, taconite will be harmful for the healing performance.

## 4. Conclusions

Polyphosphoric acid (PPA), a modifying agent, is popularly used in bitumen to enhance its elastic property. The effect of it on the healing performance was evaluated in this study. Since UV aging has a significant effect on bitumen, test samples were also handled with UV aging. Finally, we compared the healing ability of samples without or with PPA or taconite at different aging times. The results indicated that ultraviolet (UV) radiation can weaken the healing capacity of bitumen modified by polyphosphoric acid and taconite. Inversely, polyphosphoric acid and taconite fillers play a positive role on the healing capacity of modified bitumen. Based on the above findings, the dosage of 30% taconite introduced into bitumen with 1% PPA are recommended for producing a mixture with a longer service lifespan. The following are the detailed conclusions drawn from our evaluation of the resulting modified bitumen.

(1)Ultraviolet (UV) radiation can weaken the healing capacity of bitumen modified by polyphosphoric acid and taconite. The longer the aging time, the slower the decay rate of the healing ability. When adding 30% taconite content, the healing index of bitumen modified by 1% PPA were 46.81, 44.84, and 24.68% at unaged, 100 h aged, and 200 h aged conditions, respectively.(2)Polyphosphoric acid can slow down the loss of healing capacity between unaged and 100 h aged bitumen with taconite. Comparing the delta healing indexes of samples without and with PPA, the values decreased from −6.95, −18.31, 16.64, and −12.24% to −2.01, 7.68, 3.96, and −4.08% for taconite dosages of 0, 10, 20, and 30%, respectively.(3)The addition of taconite filler plays a positive role on the healing capacity of modified bitumen. When subjected to 100 h of UV aging and modified by 1% PPA, the healing index of 10, 20, and 30% taconite content introduced were increased by 5.96, 8.44, and 9.98%, respectively, compared with that without taconite.(4)Polyphosphoric acid can improve the enhancing effect of taconite on the healing capacity of bitumen. As for 100 h aged samples with 30% content taconite, the healing index with 1% PPA was increased by 4.41% compared to that without PPA.(5)After 200 h of aging, the healing property of bitumen decays fast even though the sample is modified by polyphosphoric acid and taconite.

## Figures and Tables

**Figure 1 materials-16-03333-f001:**
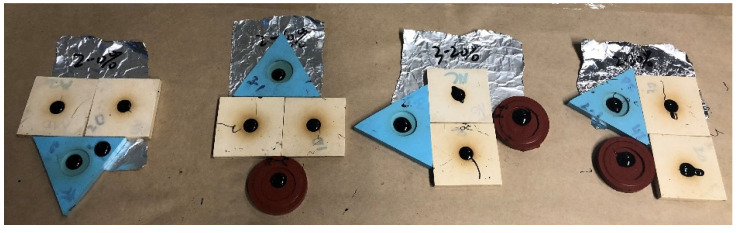
Bitumen specimens with taconite and PPA.

**Figure 2 materials-16-03333-f002:**
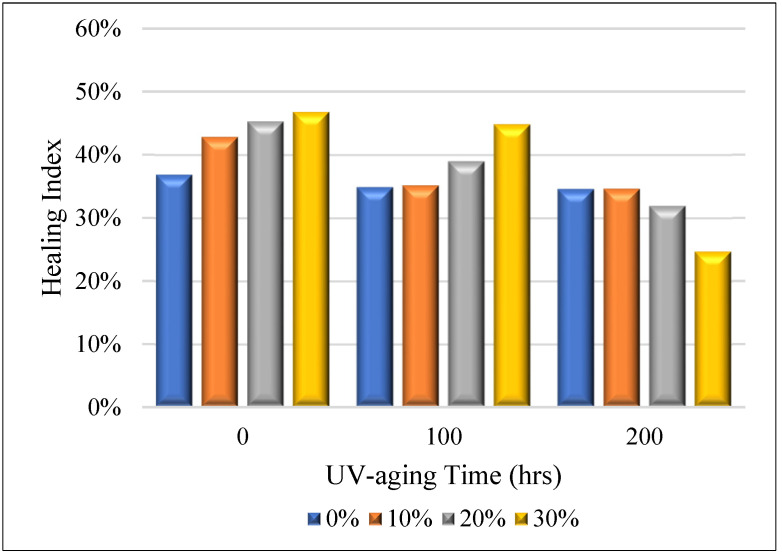
Healing indexes of bitumen modified by PPA at 0, 100 h, and 200 h UV-aging times.

**Figure 3 materials-16-03333-f003:**
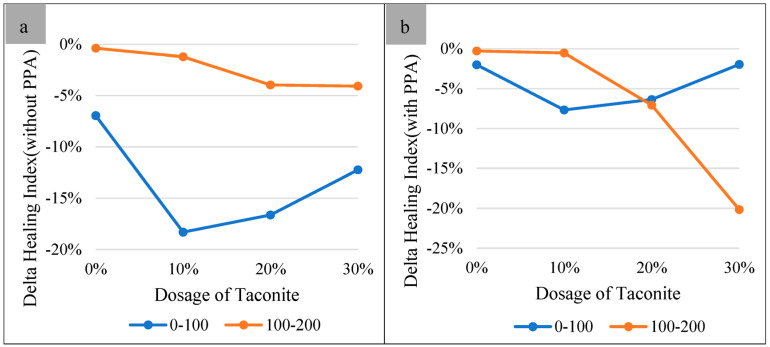
Delta healing indexes of samples during the 0–100 h and 100–200 h UV-aging periods without (**a**) and with (**b**) PPA.

**Figure 4 materials-16-03333-f004:**
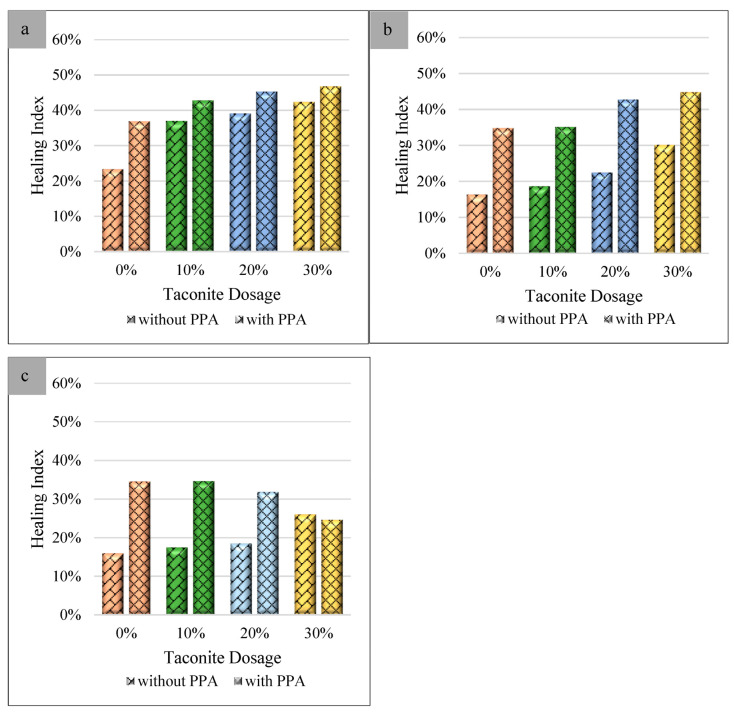
Healing indexes of samples without and with PPA at 0 (**a**), 100 h (**b**), and 200 h (**c**) of UV aging.

**Figure 5 materials-16-03333-f005:**
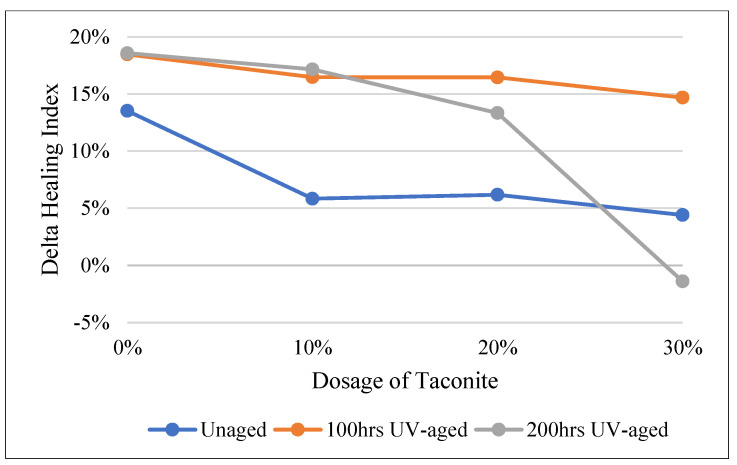
Delta healing indexes between samples without and with PPA.

**Figure 6 materials-16-03333-f006:**
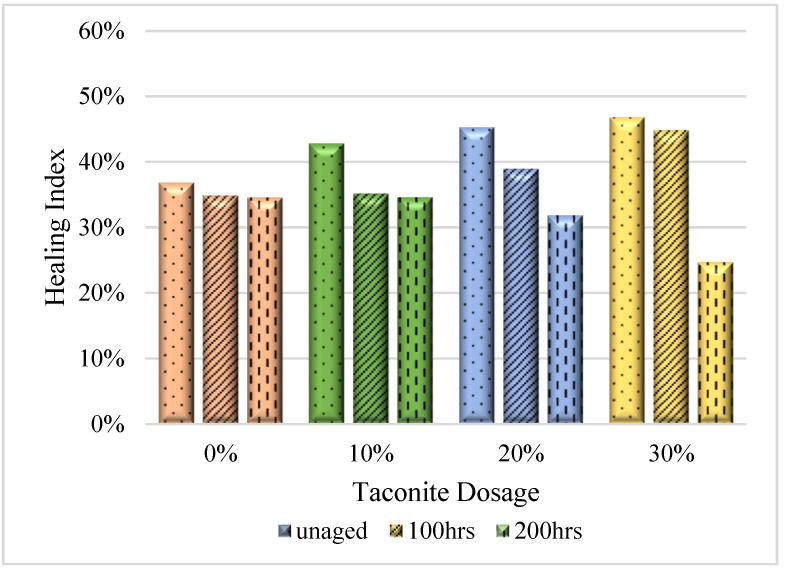
Healing indexes of PPA-modified samples with different dosages of taconite.

**Figure 7 materials-16-03333-f007:**
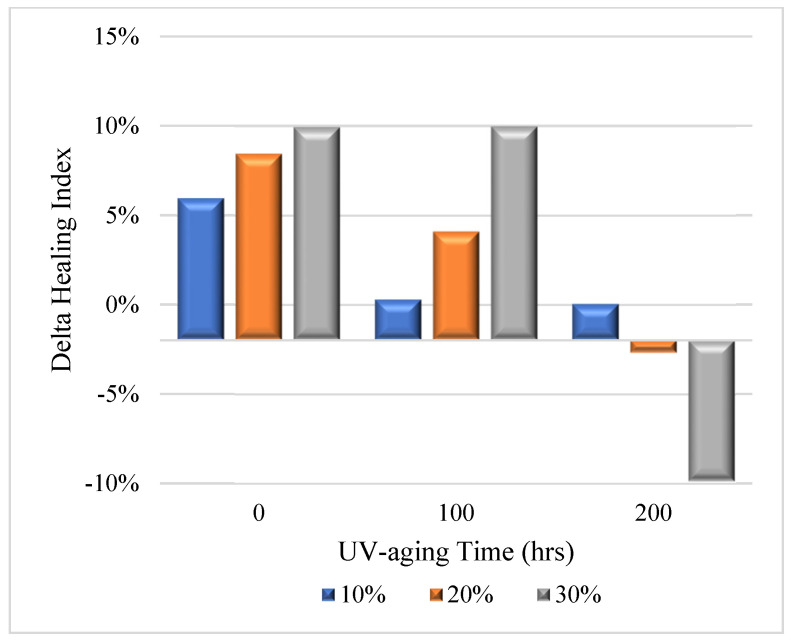
Delta healing indexes between samples without and with taconite.

## Data Availability

The data that support the findings of this study are available on request from the corresponding author.

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
