# Peer review of "Effect of Aging on Healing Capacity of Bituminous Composites Containing Polyphosphoric Acid"

_materials, 2023, doi:10.3390/ma16093333_

Round 1
Reviewer 1 Report
1- The abstract should be modified and the experiments performed should be introduced and the results should be numerically compared.
2- In the introduction, the factors affecting healing should be explained. and the effects of additives used in bitumen and on healing should be described.
3- What is the novelty of this research? Describe it at the end of the introduction.
4- How to mix bitumen and additives?
5- In section 2.2, the relevant reference or standard should be mentioned.
6- Presentation of results should be accompanied by analysis. You should analyze the relationship between the obtained results and the characteristics of the additives used.
7- In the conclusion section, the reason for achieving these results should also be explained.
8- In the conclusion section, it is better to provide a practical and executive result.
1- The abstract should be modified and the experiments performed should be introduced and the results should be numerically compared.
2- In the introduction, the factors affecting healing should be explained. and the effects of additives used in bitumen and on healing should be described.
3- What is the novelty of this research? Describe it at the end of the introduction.
4- How to mix bitumen and additives?
5- In section 2.2, the relevant reference or standard should be mentioned.
6- Presentation of results should be accompanied by analysis. You should analyze the relationship between the obtained results and the characteristics of the additives used.
7- In the conclusion section, the reason for achieving these results should also be explained.
8- In the conclusion section, it is better to provide a practical and executive result.
Author Response
1- The abstract should be modified and the experiments performed should be introduced and the results should be numerically compared.
ANSWER: Thanks for your Comments and Suggestions! We revised the abstract to the following version:
Abstract: This study examines how aging affect healing capacity of bituminous composites containing polyphosphoric acid (PPA). PPA is a commonly used in bitumen to enhance its elasticity, however, its effectiveness highly depends on its environment including internal and external factors. In terms of internal factors, the interplay between PPA and various bitumen modifiers have been extensively studied. Through ultraviolet (UV) testing and healing testing, how external factors such as exposure to ultraviolet radiation affects PPA’s efficacy measured in terms of change in bitumen’s healing index were studied. The results showed that introduction of PPA to bituminous composites significantly increases bitumen healing index, however, the change in healing index becomes less pronounced as aging progresses. Presence of other additive such as taconite found to effect efficacy of PPA with bitumen containing 30% taconite having the highest change in healing index in presence of PPA. When added with 30% taconite content, the healing index of bitumen modified by 1% PPA were 46.81, 44.84 and 24.68% at unaged, 100hrs aged and 200hrs aged condition, respectively. Overall, bitumen containing PPA had higher healing index than those without PPA regardless of extent of aging and dosage of modifiers. This in turn indicates that PPA is highly effective to enhance bitumen healing. This can be attributed to PPA promoting intermolecular interactions within bitumen matrix.
2- In the introduction, the factors affecting healing should be explained. and the effects of additives used in bitumen and on healing should be described.
ANSWER: Thanks for your comments and suggestions! We added some new content in the introduction like the following:
“……. Based on the lack of research on healing capacity of bitumen at UV-aging environmental condition and the researching direction focused currently, the effect of PPA on self-healing ability of asphalt binder with taconite was investigated for the first time and this paper also evaluated how external factors such as exposure to ultraviolet radiation affects PPA’s efficacy measured in terms of change in bitumen’s healing index. Those are attributed to understand how PPA improves capacities of bitumen totally. Different UV aging duration (0hrs, 100hrs and 200hrs), with or without PPA and different dosage of taconite were considered during the tests to evaluate those effects on healing performance. It was conducted through DSR equipment. The healing index was calculated according to the complex shear modulus. Also, this study has analyzed the promoted effect of taconite on healing performance of bitumen modified by PPA. It was found that ultraviolet (UV) radiation can weaken the healing capacity of bitumen modified by polyphosphoric acid and taconite. Inversely, polyphosphoric acid and taconite filler play a positive role on healing capacity of modified bitumen. ……”
3- What is the novelty of this research? Describe it at the end of the introduction.
ANSWER: Thanks for your comments and suggestions! We revised the introduction to the following version:
“……. Based on the lack of research on healing capacity of bitumen at UV-aging environmental condition and the researching direction focused currently, the effect of PPA on self-healing ability of asphalt binder with taconite was investigated for the first time and this paper also evaluated how external factors such as exposure to ultraviolet radiation affects PPA’s efficacy measured in terms of change in bitumen’s healing index. Those are attributed to understand how PPA improves capacities of bitumen totally. Different UV aging duration (0hrs, 100hrs and 200hrs), with or without PPA and different dosage of taconite were considered during the tests to evaluate those effects on healing performance. It was conducted through DSR equipment. The healing index was calculated according to the complex shear modulus. Also, this study has analyzed the promoted effect of taconite on healing performance of bitumen modified by PPA. It was found that ultraviolet (UV) radiation can weaken the healing capacity of bitumen modified by polyphosphoric acid and taconite. Inversely, polyphosphoric acid and taconite filler play a positive role on healing capacity of modified bitumen. ……”
4- How to mix bitumen and additives?
ANSWER: Thanks for your comments and suggestions! We added some new content in the Section 2.1 like the following:
“……. The PPA-taconite-modified binder samples were prepared using the high shear mixer to mix. The mixing temperature, time, and mixing speed was 135 °C, 30mintutes and 2000 rpm, respectively. The method of setting up the PPA-modified bitumen with taconite began at the low shear blend of 500 rpm for first 10 min, after that, the speed of blender is raised to 2000 rpm for a constant 20 min. ……”
5- In section 2.2, the relevant reference or standard should be mentioned.
ANSWER: Thanks for your comments and suggestions! For UV testing method, we referred the method which was obtained after we experienced a lot of fronting tests. Therefore, there is no relevant reference or standard.
6- Presentation of results should be accompanied by analysis. You should analyze the relationship between the obtained results and the characteristics of the additives used.
ANSWER: Thanks for your comments and suggestions! We have explained the relationship between the results and the characteristics of the additives used. Meanwhile, we also added some new explanations in the Section 3 like the followings:
“……. The healing index were weakened with the UV aging period increased. It is suggested that the effect of UV radiation on healing capacity does cause a remarkable influence. It may due to a dramatic increase in stiffness and loss of viscoelasticity after 20h of UV exposure [30]. Those changes can reduce the cracking healing property. ……”
“……. As shown in Figure.3, the healing index were promoted after adding PPA in bitumen with taconite regardless of the UV aging type. It was due to a suitable PPA gelation can improved the rheological behavior of bitumen [31]. The addition of PPA can increase the viscosity and adhesion ability of bitumen [32-33]. It will be helpful enhance the resistance of deformation, then improve the self-healing property. ……”
“……. It is suggested that the effect of UV radiation on healing capacity does cause a remarkable influence. It may due to a dramatic increase in stiffness and loss of viscoelasticity after 20 h of UV exposure [30]. More aging time longer, the bigger amount of asphaltene in the asphalt. That makes the modified binders harder and therefore less able to flow at the same temperature, which impairs its healing ability.……”
“……. It indicated that taconite can enhance the healing performance of bitumen. That’s due to the taconite would help to promote its thermal conductivity which is consistent with the results of our published paper [37]. Higher thermal conductivity can make the temperature of binders increase more quickly and get to a critical point that drives asphalt binder to start the healing behavior. ……”
7- In the conclusion section, the reason for achieving these results should also be explained.
ANSWER: Thanks for your comments and suggestions! We revised the conclusion to the following version:
“……. According to Polyphosphoric acid (PPA), a modifying agent, is very popular to used in bitumen for enhancing its elastic property, this paper evaluated the effect of it on healing performance. And since UV aging has a significant effect on bitumen, so those samples were also handled with UV aging. Finally, comparing the healing ability of without or with PPA at different aging time and the presence of taconite. The results indicated that ultraviolet (UV) radiation can weaken the healing capacity of bitumen modified by polyphosphoric acid and taconite. Inversely, polyphosphoric acid and taconite filler play a positive role on healing capacity of modified bitumen. Bases on above fingdings, the dosage of 30% taconite introducing in bitumen with 1% PPA can be considered when producing mixture. The followings are are the conclusions in detail drawn from our evaluation of the resulting modified bitumen. ……”
8- In the conclusion section, it is better to provide a practical and executive result.
ANSWER: Thanks for your comments and suggestions! We revised the conclusion to the following version:
“……. According to Polyphosphoric acid (PPA), a modifying agent, is very popular to used in bitumen for enhancing its elastic property, this paper evaluated the effect of it on healing performance. And since UV aging has a significant effect on bitumen, so those samples were also handled with UV aging. Finally, comparing the healing ability of without or with PPA at different aging time and the presence of taconite. The results indicated that ultraviolet (UV) radiation can weaken the healing capacity of bitumen modified by polyphosphoric acid and taconite. Inversely, polyphosphoric acid and taconite filler play a positive role on healing capacity of modified bitumen. Bases on above fingdings, the dosage of 30% taconite introducing in bitumen with 1% PPA can be considered when producing mixture. The followings are are the conclusions in detail drawn from our evaluation of the resulting modified bitumen. ……”
Thanks a lot for your comments and suggestions for improving our article!

Reviewer 2 Report
Reviewer’s Comments:
The manuscript “Effect of Aging on Healing Capacity of Bituminous Composites Containing Polyphosphoric Acid” is very interesting work. This paper investigates the how aging affect healing capacity of bituminous composites containing polyphosphoric acid (PPA). PPA is commonly used in bitumen to enhance its elasticity, however, its effectiveness highly depends on its environment including internal and external factors. In terms of internal factors, the interplay between PPA and various bitumen modifiers have been extensively studied. Here, we study how external factors such as exposure to ultraviolet radiation affect PPA’s efficacy measured in terms of change in bitumen’s healing index. The study results showed that introduction of PPA to bituminous composites significantly increases bitumen healing index, however, the change in healing index becomes less pronounced as aging progresses. However, the following issues should be carefully treated before publication.
1. In abstract, the author should add more scientific findings.
2. Keywords: the synthesized system is missing in the keywords. So, modify the keywords.
3. In the introduction part, the introduction part is not well organized and cited references should cite recently published articles such as 10.1016/j.reactfunctpolym.2018.07.014, 10.1039/C9RA09349D
4. Introduction part is not impressive and systematic. In the introduction part, the authors should elaborate the scientific issues in the Healing Capacity research.
5. The author should provide reason about this statement “Overall, bitumen containing PPA had higher healing index than those without PPA regardless of extent of aging and dosage of modifiers. This in turn indicates that PPA is highly effective to enhance bitumen healing. This can be attributed to PPA promoting intermolecular interactions within bitumen matrix”.
6. The authors should explain regarding the recent literature why “For unaged and 100hrs UV-aged samples, the healing index were improved gradually with the increase of dosage taconite
included”.
7. Results and Discussion. The author should explain the latest literature “To evaluate the effect of PPA on healing performance of aged bitumen with taconite, delta healing index between without and with PPA were computed (Figure. 4)”.
8. The author should provide reason about this statement, “However, when handled with 200hrs UV-aging, the healing property was reduced with the addition of taconite increased”.
9. Comparison of the present results with other similar findings in the literature should be discussed in more detail. This is necessary in order to place this work together with other work in the field and to give more credibility to the present results.
10. The conclusion part is very weak. Improve by adding the results of your studies.
Minor editing of English language required
Author Response
Answers to Comments and Suggestions
1. In abstract, the author should add more scientific findings.
ANSWER: Thanks for your Comments and Suggestions! We revised the abstract to the following version:
“Abstract: This study examines how aging affect healing capacity of bituminous composites containing polyphosphoric acid (PPA). PPA is a commonly used in bitumen to enhance its elasticity, however, its effectiveness highly depends on its environment including internal and external factors. In terms of internal factors, the interplay between PPA and various bitumen modifiers have been extensively studied. Through ultraviolet (UV) testing and healing testing, how external factors such as exposure to ultraviolet radiation affects PPA’s efficacy measured in terms of change in bitumen’s healing index were studied. The results showed that introduction of PPA to bituminous composites significantly increases bitumen healing index, however, the change in healing index becomes less pronounced as aging progresses. Presence of other additive such as taconite found to effect efficacy of PPA with bitumen containing 30% taconite having the highest change in healing index in presence of PPA. When added with 30% taconite content, the healing index of bitumen modified by 1% PPA were 46.81, 44.84 and 24.68% at unaged, 100hrs aged and 200hrs aged condition, respectively. Overall, bitumen containing PPA had higher healing index than those without PPA regardless of extent of aging and dosage of modifiers. This in turn indicates that PPA is highly effective to enhance bitumen healing. This can be attributed to PPA promoting intermolecular interactions within bitumen matrix.”
2. Keywords: the synthesized system is missing in the keywords. So, modify the keywords.
ANSWER: Thanks for your Comments and Suggestions! We revised the keywords to the following version:
“Keywords: Polyphosphoric acid; healing; modified bitumen; aging; ultraviolet”
3. In the introduction part, the introduction part is not well organized and cited references should cite recently published articles such as 10.1016/j.reactfunctpolym.2018.07.014, 10.1039/C9RA09349D
ANSWER: Thanks for your Comments and Suggestions! We have learned the recommend references in detail and found that maybe they are little related to our research which studies on bitumen modified by PPA. Our new modified material will be used on road construction. But we think those references are all much good and helpful studies.
4. Introduction part is not impressive and systematic. In the introduction part, the authors should elaborate the scientific issues in the Healing Capacity research.
ANSWER: Thanks for your Comments and Suggestions! We added some new statements in introduction like the following:
“…….As such, investigating and evaluating the effect of UV-aging on healing performance of bitumen will be helpful because it directly influencing the service life of bitumen pavement [2]. In order to enhance asphalt self-healing, many scholars have explored the healing affects by different physical methods [3-5]. Some studies have attempted to use microcapsules, hollow-fiber tubes, and nanoparticles [6-14]. ……”
5. The author should provide reason about this statement “Overall, bitumen containing PPA had higher healing index than those without PPA regardless of extent of aging and dosage of modifiers. This in turn indicates that PPA is highly effective to enhance bitumen healing. This can be attributed to PPA promoting intermolecular interactions within bitumen matrix”.
ANSWER: Thanks for your Comments and Suggestions! We added some new explanations in Section 3.2 like the following:
“…….As shown in Figure.3, the healing index were promoted after adding PPA in bitumen with taconite regardless of the UV aging type. It was due to a suitable PPA gelation can improved the rheological behavior of bitumen [34]. The addition of PPA can increase the viscosity and adhesion ability of bitumen [35-36]. It will be helpful enhance the resistance of deformation, then improve the self-healing property. ……”
6. The authors should explain regarding the recent literature why “For unaged and 100hrs UV-aged samples, the healing index were improved gradually with the increase of dosage taconite included”.
ANSWER: Thanks for your Comments and Suggestions! We added some new explanations in Section 3.3 like the followings:
“……. It indicated that taconite can enhance the healing performance of bitumen. That’s due to the taconite would help to promote its thermal conductivity which is consistent with the results of our published paper [37]. Higher thermal conductivity can make the temperature of binders increase more quickly and get to a critical point that drives asphalt binder to start the healing behavior. ……”
7. Results and Discussion.The author should explain the latest literature “To evaluate the effect of PPA on healing performance of aged bitumen with taconite, delta healing index between without and with PPA were computed (Figure. 4)”.
ANSWER: Thanks for your Comments and Suggestions! We revised the statement to the following version:
“……. To evaluate the effect of PPA on healing performance of aged bitumen with taconite, delta healing index, which represents the difference value of healing index between taconite-modified binders without and with PPA, were computed (Figure. 4). ……”
8. The author should provide reason about this statement, “However, when handled with 200hrs UV-aging, the healing property was reduced with the addition of taconite increased”.
ANSWER: Thanks for your Comments and Suggestions! We added some new explanations in Section 3.1 like the following:
“……. It is suggested that the effect of UV radiation on healing capacity does cause a remarkable influence. It may due to a dramatic increase in stiffness and loss of viscoelasticity after 20 h of UV exposure [30]. More aging time longer, the bigger amount of asphaltene in the asphalt. That makes the modified binders harder and therefore less able to flow at the same temperature, which impairs its healing ability.……”
9. Comparison of the present results with other similar findings in the literature should be discussed in more detail. This is necessary in order to place this work together with other work in the field and to give more credibility to the present results.
ANSWER: Thanks for your Comments and Suggestions! Before studying, we have reviewed a lot references and materials. There is few research on the healing performance of PPA-taconite-modified bitumen. Therefore, we just listed some related references, such as studies on other properties. But we have compared and sated the results among different modified samples to prove the benefits of PPA and taconite introduced into bitumen.
10. The conclusion part is very weak. Improve by adding the results of your studies.
ANSWER: Thanks for your Comments and Suggestions! We revised the conclusion to the following version:
“……. According to Polyphosphoric acid (PPA), a modifying agent, is very popular to used in bitumen for enhancing its elastic property, this paper evaluated the effect of it on healing performance. And since UV aging has a significant effect on bitumen, so those samples were also handled with UV aging. Finally, comparing the healing ability of without or with PPA at different aging time and the presence of taconite. The results indicated that ultraviolet (UV) radiation can weaken the healing capacity of bitumen modified by polyphosphoric acid and taconite. Inversely, polyphosphoric acid and taconite filler play a positive role on healing capacity of modified bitumen. Bases on above fingdings, the dosage of 30% taconite introducing in bitumen with 1% PPA can be considered when producing mixture. The followings are are the conclusions in detail drawn from our evaluation of the resulting modified bitumen. ……”
Thanks a lot for your comments and suggestions for improving our article!

Reviewer 3 Report
The reviewer thanks the authors and editors for the opportunity to review the manuscript. The article discusses issues the effect of aging on healing capacity of bituminous composites containing polyphosphoric acid and tacomite
The article is interesting and may provide a basis for considering the extension this issue. Nevertheless, it should be emphasized that the presented methods, test results and results discussion need in depth information and are insufficient. Also, further analysis should still be conducted.
All my comments are highlighted in the PFD reviewed version.
Extensively English revision is needed.

All my comments are highlighted in the PFD reviewed version.
Extensively English revision is needed.
Author Response
Please see the PDF file. We made all answers to the each of comments from Reviewer 3.

Round 2
Reviewer 1 Report
The changes made are approved
The changes made are approved
Reviewer 2 Report
Accept in present form
Minor editing of English language required
Reviewer 3 Report
Accept in the present form with minor editing of English language required.
Accept in the present form with minor editing of English language required.